# Priorities and barriers for ageing well; results from stakeholder workshops in rural and urban Rwanda

**Theogene Uwizeyimana**[1], **Aimable Uwimana**[1], **Collins Fred Inkotanyi**[1], **Dina Goodman-Palmer**[2]*, **Samuel Ntaganira**[1], **Leslie Kanyana**[1], **Maria Lisa Odland**[2,3], **Sandra Agyapong-Badu**[4], **Lisa Hirschhorn**[5,6], **Tsion Yohannes**[1], **Carolyn Greig**[4,7,8‡], **Justine Davies**[2,6,9‡]

1 University of Global Health Equity, Kigali, Rwanda, 2 Institute of Applied Health Research, University of Birmingham, Birmingham, United Kingdom, 3 Norwegian Technical University, Trondheim, Norway, 4 NIHR Birmingham Biomedical Research Centre, University Hospitals Birmingham NHS Foundation Trust and University of Birmingham, Birmingham, United Kingdom, 5 Robert J. Havey, MD Institute for Global Health—Ryan Family Center for Global Primary Care, Northwestern University, Chicago, IL, United States of America, 6 Department of Global Health, Centre for Global Surgery, Stellenbosch University, Cape Town, South Africa, 7 School of Sport, Exercise and Rehabilitation Sciences, University of Birmingham, Birmingham, United Kingdom, 8 MRC-Versus Arthritis Centre for Musculoskeletal Ageing Research, University of Birmingham, Birmingham, United Kingdom, 9 Medical Research Council/Wits University Rural Public Health and Health Transitions Research Unit, Faculty of Health Sciences, School of Public Health, University of the Witwatersrand, Johannesburg, South Africa

‡ CG and JD are joint senior authors on this work.
* dina.goodman@gmail.com

**Data Availability Statement:** All relevant data are within the paper and its Supporting Information files (specifically S1 Fig).

## Abstract

### Background

The National Older Person's Policy of 2021 in Rwanda highlights the need for social protection of older populations. However, there is a lack of local knowledge regarding the priorities and challenges to healthy aging faced by older people and their caregivers.

### Objectives

This study aimed to identify and compare the needs and priorities of older people and other stakeholders involved in caring for them in rural and urban areas of Rwanda.

### Methods

The study was conducted in two locations, Kigali (urban) and Burera district (rural). Each site hosted two separate one-day workshops with older people ($\geq$60 years) and stakeholders (all ages). Discussions were held in plenary and roundtable-groups to generate a list of the top 4 prioritized responses on areas of importance, priorities/enablers to be addressed, and obstacles to living a healthy and active life for older people. The research team identified similarities between stakeholder and older people's responses in each area and a socio-ecological model was used to categorize findings.

**Funding:** JD and CG were funded by the Institute of Global Innovation (IGI) at the University of Birmingham, UK. The funding was internal, so there is no grant number. The funders had no role in study design, data collection and analysis, decision to publish, or preparation of the manuscript. The URL of the IGI website is: https://www.birmingham.ac.uk/research/global-goals/igi/index.aspx.

**Competing interests:** The authors have declared that no competing interests exist.

**Abbreviations:** NISR, National Institute of Statistics of Rwanda; MINECOFIN, Ministère d'Economie et Finance or Ministry of Finance and Economic Planning; MOH, Ministry of Health; UGHE, University of Global Health and Equity; IRB, Institutional Review Board; VUP, Vision Umurenge Program; ICT, Information and Communications Technology; LMIC, Lower- and Middle-Income Country; MINALOC, Ministry of Local Government; RSSB, Rwanda Social Security Board; JADF, Joint Actions Development Forum; PSF, Private Sector Federation; BDH, Butaro District Hospital; IMB, Inshuti Mu Buzima; POSER, Program on Social and Economic Rights.

## Results

There were substantial differences in responses between rural and urban areas and between older people and stakeholders. For each question posed, in each rural or urban area, there was only agreement between stakeholders and older people for a maximum of one response. Whereas, when comparing responses from the same participant groups in urban or rural settings, there was a maximum agreement of two responses, with two questions having no agreement in responses at all. Responses across all discussion-areas were mostly categorized within the Societal level, with Individual, Relationship, and Environment featuring less frequently.

## Conclusion

This study highlights the need for contextually curated interventions to address the concerns of older adults and their caregivers in rural and urban settings. An inclusive and multidimensional approach is needed to conquer the barriers that impede healthy aging, with input from various stakeholders.

## Introduction

Rwanda, a small landlocked country, is one of the most densely populated countries in Africa with an estimated population of 12 million. Approximately 10% of the population are over 50 years old, while the old-age dependency ratio (i.e., the number of people 65 years per 100 people 15–64 years) is 5.3 [1]. The most recent data from the headline results of the Population and Housing Census of Rwanda of 2022 indicates that life expectancy in Rwanda has significantly increased from 53.7 in 1991 to 69.6 years in 2022 [2]. This is expected to increase to 71.4 years by 2027 [3]. Projections indicate that the size of the population of older people in Rwanda will double between 2012 and 2032, with higher growth in urban areas [1, 4, 5].

Recognizing the need for social protection of older populations, Rwanda has developed a National Older Person's Policy (NOPP) of 2021, that emphasizes the need to focus on emerging needs of older populations. These include social security policies and programs for populations engaged in the informal system [6]. According to this policy, older persons are defined as people 65 years of age or older, and have experienced a change in their social roles, "functional, psychological, and intellectual abilities" [6].

The original NOPP was developed in Rwanda in 2016 [6]. It is a comprehensive policy that outlines the government's commitment to the well-being of older persons. It aims to promote the rights and dignity of older persons, to ensure their access to essential services, and to support their participation in society. The policy was developed through a participatory process that involved a wide range of stakeholders, including older persons themselves, their families, civil society organizations, and government agencies. The process included public hearings and consultations with older persons and their advocacies. It has been implemented partly. Some of the key provisions of the policy, such as the establishment of a National Council for Older Persons and the provision of social protection for older persons (VUP), have been implemented. However, other provisions, such as providing long-term care services and promoting intergenerational solidarity, have not yet been fully implemented.

Agriculture is the main source of income in rural areas [7]. Furthermore, the National Census 2014–15 indicated that individuals over 60 years of age are engaged in informal jobs in the

agricultural sector [8]. This 2014–2015 census demonstrated that older women are more likely to work in agriculture than younger women, as indicated by the fact that 87 percent of women aged 45–49 and 72 percent of those aged 15–19 work in agriculture. Similarly, the majority of men work in agriculture are older (67.3 percent of men aged 45–49 as opposed to 60.4 aged 15–19) [8]. This report also indicated that the older population with health problems are in need of access to improved structures, water sources, electricity, and other energy sources [8].

Despite political will in Rwanda, there is a lack of up-to-date local knowledge around the priorities and challenges of healthy ageing among older people or the stakeholders who may care for them, or whether these are experienced differently between people in urban or rural populations. The World Health Organization defines healthy ageing as "a process of maintaining functional ability to enable wellbeing in older age." [9] This study aimed to identify and compare the needs and priorities for the achievement of healthy ageing of older people and stakeholders involved in caring for them in rural and urban areas of Rwanda. To achieve this, we rely on the philosophy of determinism, which suggests that the conditions and experiences of persons–including older adults in Rwanda—are primarily shaped by external societal and environmental factors, rather than individual agency alone [10]. To further categorize and analyze our findings, we use the socio-ecological model (SEM) as a framework. This model allows for a comprehensive understanding of the multiple influences on an individual's health and well-being, including Individual, Relational, Environmental, and Societal factors [11].

## Methods

### Study setting

This study was carried out in two locations in Rwanda: (1) Kigali, an urban area and (2) Burera district, a rural location. Kigali is the capital and largest city of Rwanda with a population of almost 2 million total, with 30,000 inhabitants over 60 years of age. Burera district is located in the rural northern province of Rwanda and has a population of 350,000, with 5.6% of the total population over 60 years. Agriculture is the main source of income for people in the Burera district, while the service sector is the primary source of income in Kigali.

These areas were selected to highlight the differences between urban and rural populations in Rwanda, whilst also being feasible to study. In particular, the University of Global Health Equity (UGHE) is situated in Burera district and has community engagement programs allowing us to bring together older people for the workshops in a remote rural area who would ordinarily be hard to access.

### Study population

This study recruited two groups of participants; older people (60 years and older) and stakeholders (of all ages) who work closely with older people. The study team chose to include individuals 60 and older rather than 65 and older due to life expectancy (69.6 years in Rwanda in 2022) in the context of this study. Stakeholders were purposively selected from a broad range of organizations (i.e., charitable, civil society, government and religious) that are involved in providing services or care for older people. For each stakeholder type, at least 1–2 representatives were invited. Contacts of leaders of these organizations were obtained from a list held at local district administration centres. Older people were then identified by the stakeholders. Stakeholders were requested to nominate older participants to represent a broad range of characteristics (e.g., those living with disabilities and chronic diseases, farmers, and those who worked in public or private services before retirement).

The nominated elderly people were then listed in random order by district and cell (a subdivision of a sector, which is a subdivision of a district), and were called by phone until the

required number confirmed their availability. From each cell at least 2–3 people were selected such that a total of around 15 elderly individuals were invited in each study area (Kigali and Burera). The aims of the workshops and different planned activities were explained to all potential participants of the workshops by a local researcher through a phone call. Participants who expressed interest were given the details of the workshop time and location and provided with transport to attend. All communication was conducted in Kinyarwanda, the local language, and results were translated into English for analysis. Translations into English were done by the Rwanda team, and both Rwandan and English-speaking authors then discussed the translation to ensure that the meaning was clear in English and remained accurate in Kinyarwanda.

## Ethics and consent

Ethical approval for the study was given by the University of Global Health Equity (UGHE) Institutional Review Board UGHE-IRB/2021/018 and the University of Birmingham Science Technology Engineering and Mathematics Research Ethics Committee ERN_21–0632. Written informed consent was obtained from all literate participants and oral witnessed informed consent for all others. Participants were provided with transport, lunch, and refreshments, but there were no financial incentives to attend. All participants were tested for COVID-19 prior to workshops taking place.

## Data collection procedure

Each study site hosted two separate 1-day workshops: one day for older people and another for stakeholders. Workshops took place at convenient locations in the communities (Butaro sector hall for Burera district and a centrally-located hotel in Kigali city). The nominal group technique was used as a method for engaging discussions within groups and getting rapid agreement [12]. This method is consistent with our determinist approach, as it allows for the articulation of the influence of external societal and environmental factors on the individual, relational, environmental, and societal perspectives of both older people and stakeholders.

The first workshop was held with stakeholders and proceeded as follows. After an introduction explaining the rationale and proceedings, stakeholders were allocated into small roundtable discussion groups, each with a facilitator, to discuss following topics in turn: (1) "What do you think is important for older people in Rwanda?" and (2) "What are the main priorities that need to be addressed to maintain health and wellbeing for older people in Rwanda?". Facilitators captured a list of all responses from their discussion group. When responses from the discussion groups were unclear, facilitators worked with the discussion group to improve clarity. Towards the end of the small group discussions, the facilitator asked group members to review the list and prioritise their top 4 responses in order of importance. After this, one member from each group presented the ranked top 4 priorities from their group to the whole group in plenary, allowing opportunity for all members of the plenary group to discuss the rational for the selected responses and their priority order. Following the discussion, the facilitators combined the top 4 responses from each of the small groups after discussing amongst the facilitator group which were duplicate responses and removing these. All stakeholders then voted for the top 4 priorities for each question; facilitators recorded these priorities.

The nominal group technique focuses on the nomination of ideas by participants at roundtables. Discussion was encouraged in the roundtable and plenary sessions prior to the final ranking of the lists, such that the ranking was informed by the prior discussion. Hence, disagreements on the selected priorities or their orders are uncommon. Where disagreements occurred, facilitators worked with the discussion group to improve agreement. The final list

was selected by voting by all participants, and disagreement on the order after voting is unusual in nominal group technique methods.

Finally, in order to get a sense of met need and awareness, stakeholders were asked to list all "services, and family and community structures available to ensure that older people are able to live healthy active lives in Rwanda." These discussions were also conducted in small groups with the complete list presented in plenary and discussed with an aim to ensure that all services were captured.

The format for the second workshop, with older people, was similar to that for stakeholders with some small differences—for ease of comprehension—in the topics discussed and prioritized by small groups. These were (1) "What do you think is important for older people in Rwanda?", (2) "What are the main enablers to ensure that you live a healthy and active life?", and (3) "What are the main obstacles to ensure you are living a healthy and active life?" As for the stakeholder group, after each question was discussed, one person from each small group presented the ranked findings from their discussions to the plenary. There followed a whole group discussion at the end of which all the older participants voted for their top 4 priorities for each question. At the end of the voting, the rankings were recorded by the investigators. While topic questions were slightly different, priorities were identified using the same process for older people as for the stakeholders.

The facilitators did not ask their round-table group to focus on any individual's reported priorities, rather the facilitators supported their group in their selection of consensus priorities that the whole group felt most adequately represented their choice. TU, AU, LK and SN facilitated the workshops.

All reported responses were single answers provided by the groups after they had agreed on the priority list. These workshops were not qualitative focus groups, so they were not audio recorded or transcribed verbatim and coded.

## Analyses

Stakeholder and older people's responses were summarised. After which, the research team met to discuss and develop consensus on the similarities between stakeholders and older people's responses in each discussion-area, considering the top four ranked priorities from each plenary discussion. Similarities between groups are shown in figures.

We further categorized responses based on the Socio-Ecological Model (SEM), which is a framework advanced by the World Health Organization. It aids in understanding the diverse influences on an individual's health and well-being, including levels such as Individual, Relationships, Institutions, Social or Community, Environmental, and Policy [11]. We used four levels selected based upon the responses received: 1. Individual—pertaining to the internal attributes of individuals, for example, their demographics or health; 2. Relationship–pertaining to relationships proximate to the individual, for example with their family and friends; 3. Environment–being the surroundings in which person lives, including its organisations, services, and structures; and 4. Society–pertaining to the culture, social, and policy (including health, legal, and economic) milieu in which the person lives [13]. Where possible, each response was categorized into one of these levels, though more than one level of the socio-ecological framework could be used if a single response was best represented as such. We recognize the dynamic nature of components of SEM but for the purposes of this study we considered each discreet response to be categorised. This was agreed upon discussion between the authors.

As described above, we were guided by philosophy of determinism, for example, for a person to achieve their needs there needed to be provision of externalities to enable that. Hence, for example, an individual's ability to eat or to do sport were considered primarily a factor of

the environmental or societal provision of enablers. We categorized things that are experienced by the individual, such as their own health, as 'individual'. The first 30% of responses were categorised by CG & JD, together; the subsequent 10% of responses were categorised by each separately until 100% agreement had been achieved (which occurred after the first 10%); after-which the remaining responses were categorised by JD.

## Results

Workshops were held between 24 and 25 of June 2021 in Kigali, and 06 and 07 of October 2021 in Burera. In Kigali, 13 stakeholders and 15 older people were invited to participate and, in Burera, 15 stakeholders and 15 older people were invited. For the workshops with stakeholders 10 (5 male and 5 female) attended in Kigali and 13 (10 male and 3 female) in Burera. A full list of stakeholder organizations that were invited is given in S1 Table. For the workshops with older people 12 (8 male and 4 female) attended in Kigali and 15 (8 male and 7 female) in Burera.

The top four stakeholder and older persons' responses are summarized in Table 1. All responses are presented in S1 Fig. The results from both workshops highlighted that it is important to older people to prioritise physical and mental health, improve the health system and provide access to healthcare services, and provide safe housing. Stakeholders in both workshops felt it was important for older people to have a healthy and close-knit family, be financially stable, live in a peaceful country, preserve culture and tradition and own properties. The priorities that need to be addressed, put forwards by stakeholders, include creating advocacy for the elderly at a national level, establishing an elderly monitoring office, the provision of a healthcare insurance scheme and a pension plan, and being cared for by government and family. The main enablers that older people put forwards to help them maintain health and well-being were good governance and security, improved local infrastructure, good nutrition, and ability to pay for medical services. The main obstacles that older people put forwards were difficulty accessing transport services, lack of food markets specifically for older people, inability to meet their basic needs due to poverty, and chronic illness.

For each question, responses differed between rural and urban areas and between older people and stakeholders (**Figs 1 and 2**). For example, when contrasting responses between older people and stakeholders living in the same area, for Kigali stakeholders (**Fig 1**), when considering what is most important for older people in Rwanda, the most highly ranked response related to having a close-knit family and financial stability. Whereas for older people in Kigali, health and wellbeing was prioritized (health was the fourth ranked priority from stakeholders in Kigali). There was slightly more overlap in the top 4 responses between stakeholders and older people living in Burera where there was a shared need expressed for housing and medical insurance. Whereas there was only overlap in one priority in Kigali where both older people and stakeholders mentioned need for better economic support (including for health)

Similarities between the same type of respondents living in different areas are shown in **Fig 2**. There was little overlap between stakeholder-ranked priorities that need to be addressed to maintain health and well-being and no overlap at all between obstacles to ensuring older people live a healthy and active life listed by older people in both Kigali and Burera.

Responses categorized using the socio-ecologic model are shown in **Fig 3**. Across both groups and locations, issues of importance were mostly categorized within the Societal level, with Individual, Relationship, and Environment featuring less frequently. When considering priorities which needed to be addressed (stakeholders) and the obstacles and enablers (older people), in Kigali, all issues ranked highly by stakeholders were classified within Society,

**Table 1. Top four priorities responses from older people and stakeholders in Burera and Kigali (see appendices for all listed responses).**

| What do you think is important for older people in Rwanda? | | | | What are the main priorities that need to be addressed to maintain health and wellbeing for older people in Rwanda? | | What are the main enablers to ensure that you live a healthy and active life? | | What are the main obstacles to ensure you are living a healthy and active life? | |
|---|---|---|---|---|---|---|---|---|---|
| Kigali Stakeholders | Kigali Older People | Burera Stakeholders | Burera Older People | Kigali Stakeholders | Burera Stakeholders | Kigali Older People | Burera Older People | Kigali Older People | Burera Older People |
| Having a healthy and close-knit family that is stable financially | Prioritizing physical/ mental health and social wellbeing | Preserve traditional culture and practices | Access to healthcare services | Creating elderly advocacy on a national level | Provision of health care insurance scheme and pension plan | Good governance and security which allows a peaceful, happy and safe life with their family and friends | Good nutrition (balanced diet) | Difficulty accessing public transport services | Unable to meet basic needs of daily life (e.g: hygiene and self care) living due to poverty |
| Living in a safe, peaceful and economically stable country | Improving the public health system while receiving routine medical care (primary care) and special medical care (geriatric) | Owning properties | Safe houses to live in that are free from violence and structurally safe | Establishing an elderly monitoring office involving different stakeholders | Being cared for (support from government, family, society) | Improved local Infrastructure (e.g.: paved roads, electricity and water supply, internet, etc.) which allows the elderly to be connected to society and to their peers | Ability to pay for medical services (having a health insurance scheme) | Lack of common food markets specifically for older people | Chronic illness |
| Having time and space for religious and spiritual needs | Lobbying and advocacy at all levels in the government by representative members of the elderly community | Caring for their families | Self-respect | Establishing a disaggregated database of the elderly to better address their needs | provision of adequate shelter/ housing for living | Providing financial and health assistance through various programs (e.g. Mutuelle de santé, pensions, Ejo Heza, Subsidy/Vision Umurenge Program (VUP)) | Having a good relationship with neighbors | Access & quality challenges related to accessing medical care services | Conflicts within the family |
| Maintaining physical and mental strength | Providing to the needy elderly people a financial support | Caring for their own life by having good hygiene and access to healthcare | Living in a secured neighborhood and country | Better structured socio- economic welfare support system | Being engaged in regular activities like sports | Incorporating Information and Communication Technology services and devices in businesses and economy and in their lives in general to improve communication | Living in clean and safe houses | Lack of social/ communal spaces that encourage interactions with others (playground, social areas, parks, etc) | Lack of caretakers |

whereas all obstacles defined by older people were mostly Environmental, although enablers spanned categories of Relationship, Environment, and Society; there were no enablers or obstacles which were categorized as Individual. In Burera, Stakeholders' ranked priorities that needed to be addressed in more categories than in Kigali; with those in Relationship, Environment, and Society being mentioned. Obstacles and enablers prioritized by older people in Burera spanned all levels of the Socioecological Framework.

## Discussion

In this description of workshop findings to explore the priority-ranked issues of importance for older people in Rwanda, the priorities that need to be addressed to enable health and wellbeing, and the obstacles and enablers to ensure older people live a health active life, we found

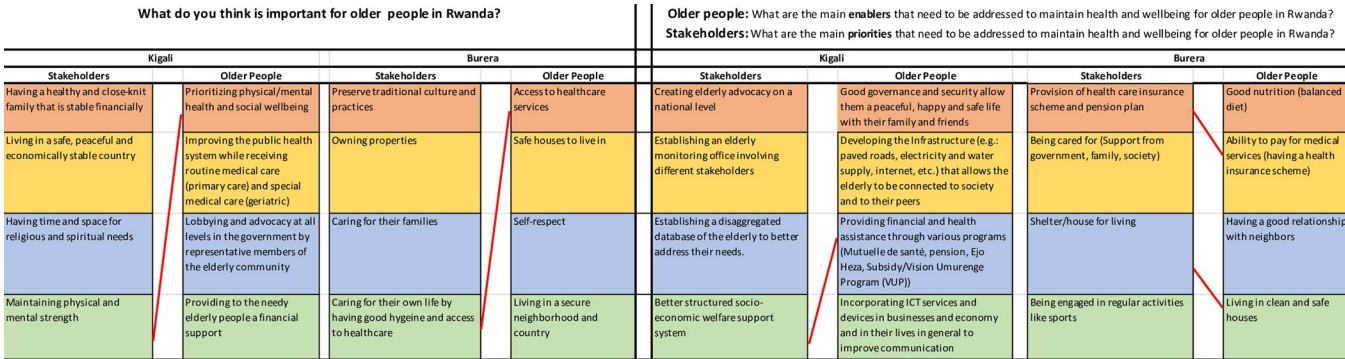

**Fig 1. Top ranked responses for each question/theme with red connecting lines indicating agreement in responses between stakeholders and older people from each geographical area.**

differences in priorities between older people and stakeholders and further differences between groups dependent on geography.

When considering areas of importance to older people, a mix of responses were mentioned by all groups, spanning all levels of the socio-ecological framework [13]. Given that older people are considered likely to suffer health issues, it is of interest that domains intrinsic to individuals were mentioned infrequently and only in two instances were these directly related to health (e.g: maintaining physical and mental health, and chronic illness). Most responses were categorized by both older people and stakeholders, in both geographical locations, in the Society level, suggesting a shared perception of importance of the culture, social, and policy milieu in which older people live. However, when comparing the actional areas listed, there was little agreement between older people and stakeholders from the same geographical area, suggesting a disconnect between perceived and lived experience of areas of importance.

There was more overlap between older people and stakeholders when considering all reported areas of importance–not just the top 4 prioritized ones. Nevertheless, that stakeholders–who are likely to be in positions of more power than older people, and thus have a greater voice to speak for older people–prioritise different areas of importance to older people might mean that older-people are misrepresented in discussions where their stakeholders act as proxies for them. This disconnect between stakeholders and older people is further seen in responses to the priorities which need to be addressed–as was asked of stakeholders–and the enablers and obstacles–as was asked of older people–to maintain or ensure health and well-being for older

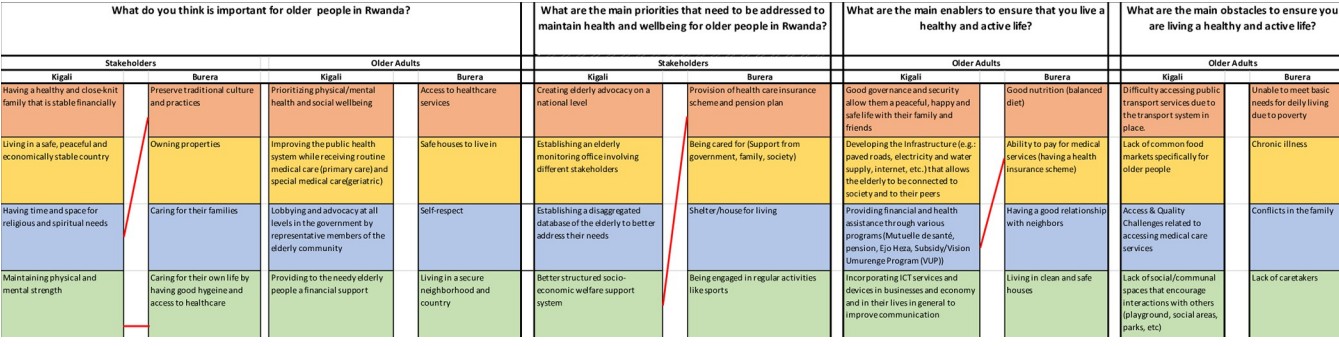

**Fig 2. Top four responses for each question with red connecting lines indicating agreement in responses between geographical area in each study group (among older people and stakeholders).**

| What do you think is important for older people in Rwanda | | | | What are the main enablers to ensure that you live a healthy and active life? | | What are the main priorities that need to be addressed to maintain health and wellbeing for older people in Rwanda? | | What are the main obstacles to ensure you are living a healthy and active life? | |
| Kigali | | Burera | | Kigali | Burera | Kigali | Burera | Kigali | Burera |
| Stakeholder | Older People | Stakeholder | Older People | Older People | | Stakeholders | | Older People | |
| Having a healthy and close-knit family that is stable financially | Prioritizing physical/mental health and social wellbeing | Preserve traditional culture and practices | Access to healthcare services | Good governance and security allow them a peaceful, happy and safe life with their family and friends | Good nutrition (balanced diet) | Creating elderly advocacy on a national level | Provision of health care insurance scheme and pension plan | Difficulty accessing public transport services due to the transport system in place | Unable to meet basic needs for daily living due to poverty |
| Living in a safe, peaceful and economically stable country | Improving the public health system while receiving routine medical care (primary care) and special medical care (geriatric) | Owning properties | Safe houses to live in | Developing the Infrastructure (e.g.: paved roads, electricity and water supply, internet) that allows the elderly to be connected to society and to their peers. | Ability to pay for medical services (having a health insurance scheme) | Establishing an elderly monitoring office involving different stakeholders | Being cared for (Support from government, family, Society) | Lack of common food markets specifically for older people | Chronic illness |
| Having time and space for religious and spiritual needs | Lobbying and advocacy at all levels in the government by representative members of the elderly community | Caring for their families | Self-respect | Providing financial and health assistance through various programs (Mutuelle de santé, pension, Ejo Heza, Subsidy/Vision Umurenge Program (VUP)) | Having a good relationship with neighbors | Establishing a disaggregated database of the elderly to better address their needs | Shelter/house for living | Access & Quality Challenges related to accessing medical care services | Conflicts in the family |
| Maintaining physical and mental strength | Providing to the needy elderly people a financial support | Caring for their own life by having good hygiene and access to healthcare | Living in a secured neighborhood and country | Incorporating ICT services and devices in businesses and economy and in their lives in general to improve communication. | Living in clean and safe houses | Better structured socio-economic welfare support system. | Being engaged in regular activities like sports | Lack of social/communal spaces that encourage interactions with others (playground, social areas, parks, etc) | Lack of caretakers |

Legend:  Individual  Relationship  Environment  Society

**Fig 3. Top ranked responses for stakeholders and older people in Kigali (urban) and Burera (rural) categorised using a socio-ecological framework approach.** Note that some responses utilise more than one domain of the socio-ecological framework.

people. Again, confirming the need to include older people in conversations around planning, policy, and action if older people are to achieve a future that is optimal for them.

A disconnect between service providers and beneficiaries is becoming a commonly observed in many studies. For example, a study done in the United States to assess the understanding of the term "obesity" among older people and their clinicians has revealed that although there is a perception amongst clinicians that older people are less affected by obesity stigma than young people, this was not the case [14]. Furthermore, a similar study done to compare patients and providers' perception of health care delivery in the US has identified a disconnect between what the healthcare provider can offer and patients' expectations [15]. However, such disconnects in perceptions haven't before been shown when considering needs of older people in a sub-Saharan African setting. Given the rapidly ageing population in sub-Saharan Africa and the urgent need to develop services and infrastructure to cater for this populations' needs, the identified disconnect between older people and stakeholders in our study informs a need to include older people in policy making, planning and action. A review done to assess the extent of literature on the participation of senior citizens (older people) in the designing of policy and their implementation in European member countries has identified different ways and initiatives to engage older people in the decision-making process [16]. This study by Falanga et al has identified three emerging patterns that include participatory initiatives that adopted methods ranging from consultative approaches in policy design and implementation, to a co-decisional approach in policy-making [16]. These initiatives can be used by many other countries, including Rwanda, to ensure that older people's needs, and priorities are included in the policy designing and implementation.

To categorise our results, we applied the socio-ecological framework and the concept of Determinism, considering that individuals have limited agency and ability to address their

own needs whereas structural and political determinants play a larger role [10]. Hence many of the ranked priorities that need to be addressed and the enablers put forwards by both stakeholders and older people were in the Societal and Environmental categories. Although individuals may have agency to improve their financial status and ability to access services for health, socialisation, and leisure, in the low-resourced setting of Rwanda it is likely that this agency is limited by a lack of resources and that the approach of Determinism is appropriate. For example, it is well understood now that diet is largely determined by Society rather than the Individual due to complex food systems [17]. Similarly, a study done in Namibia showed an environmental barrier to assessing healthcare due to long distance, something that is more common in many LMICs [18].The difference in priorities between older people and stakeholders living in rural vs. urban areas further explain how the approach of Determinism is well to this study.

Our study found clear differences between the perspectives of participants in rural and urban areas. Notably, there was more overlap between rural and urban stakeholders than there was between rural and urban older people. In fact, the only shared theme between older adults in the two study sites related to health insurance as an enabler to living a healthy active life. A similar divergence in priorities between geographical areas and stakeholder groups was found in other studies in Pakistan and Brazil (Lakhdir MPA, personal communication, June 2023 and Feriol E, personal communication, December 2023).

All study participants over 30 years of age lived through the 1994 Genocide against the Tutsi. Likely as a result, issues around safety and security were brought up in all workshops. However, these were not as frequent as expected, despite the impact that the genocide had on the study population. A report on population ageing in Rwanda shows challenging living situations among older people because of having lost or separated from their family members during the genocide, resulting in the deterioration of family structures and the lack of intergenerational support [19]. This would lead to emotional and financial insecurity because many older persons live by themselves with no one to care for them or are expected to take care of their grandchildren whose parents have died or migrated due to the Genocide [20]. However, there has been implementation of social support and capacity building programs such building houses and access to palliative care for older Genocide survivors [21] and the establishment of the NOPP aims to ensure they are involved in events and programs that improve their wellbeing socially, culturally, politically, and economically [6]. Our workshops extend beyond the NOPP. For example, not only is the NOPP and the consultation that informed it not up-to-date, the NOPP does not explicitly address the needs of older people in rural areas, nor does it show the disconnect between the priorities of older people and the priorities of stakeholders. Older people in Rwanda have the right to health and are supported in acquiring medical insurance like *Mutuelle de Sante*, receiving social support (VUP), and accessing the pension system through the Rwanda Social Support Board (RSSB). These elements were considered existing externalities in our classification using the socio-ecological model. Further studies should be done to understand the impact of capacity building interventions that support safety and wellbeing of the elderly community in Rwanda [22]. However, taken together, our findings suggest that the process that resulted in the NOPP may require to be updated with more consideration given to the balance of older people and stakeholders to ensure that the needs of older people are represented adequately.

These workshops provided opportunities to build upon a small body of literature of aging in Rwanda. Furthermore, the approach used for the workshops to build consensus allowed us to capture a variety of perspectives, both among older people and stakeholders living in urban and rural settings. Along with these aforementioned strengths, this study does have limitations to consider. First, stakeholders and older people were asked slightly different questions, which

may have led to some of the disconnects in responses reported above. However, this was done after engagement with communities to ensure that questions were understood appropriately and is likely not a major reason for differences observed. Also, these workshops did not explore effects of gender. While present-day Rwanda is considered fairly equal society in terms of gender, historically this may have been different, as indicated by the patriarchal power structure of the past with power imbalances between women and men [23, 24]. As a result, older women–like those included–may have suffered gender-based inequalities that were not captured in this study. Also, the wording of the questions may have led the results to focus on the Societal domain, however this aligns with our tendency towards a deterministic approach that sees an individual as only able to achieve their needs with the provision of enabling externalities. Despite the questions being discreet, participants could have interpreted them as similar questions and this may have created overlap between responses for each question. It is challenging in practice to know the balance between whether behavior is medicated by individual's agency or externalities. The philosophy of determinism suggests that most events or behavior are governed by externalities. This philosophy has gained traction in individual health, especially in high income countries and in particular around making healthy choices. Drivers of choices in Rwanda, whether predominately through agency or externalities has been understudied. However, on discussion, the authors agreed that in many cases, the existence of externalities was more likely to influence decisions and choices than agency for the issues under discussion. Finally, as time and resources limited data collection to a convenience sample in one urban and one rural area, these results may not be representative of all rural areas and urban areas in Rwanda.

## Conclusion

These workshops in Rwanda found differences in priority-ranked issues relating to aging well between older people and stakeholders and further differences between groups dependent on geography. This work can serve as a guide for researchers and policymakers to plan future interventions to facilitate healthy ageing in the LMIC settings. Specifically, the information from our study can be used to improve the existing NOPP in Rwanda to address the needs of older people throughout Rwanda, but especially in rural areas. These results demonstrate the need for contextually curated interventions to address the concerns of older adults and their caretakers in rural and urban settings. An inclusive and multidimensional approach is needed to conquer the barriers that impede healthy ageing, with input from various stakeholders including NGOs, public and private sector institutions, the older adult population, and their caretakers.

## Supporting information

**S1 Table. Stakeholder organizations invited.** A full list of stakeholder organizations that were invited to participate in the workshops.
(DOCX)

**S1 Fig. Ranked responses from workshops.**
(DOCX)

**S1 Document. Inclusivity in global research questionnaire.**
(DOCX)

## Acknowledgments

The authors thank Professor Janice Thompson, University of Birmingham, for her helpful advice regarding data analysis.

## Author Contributions

**Conceptualization:** Theogene Uwizeyimana, Aimable Uwimana, Collins Fred Inkotanyi, Dina Goodman-Palmer, Maria Lisa Odland, Sandra Agyapong-Badu, Tsion Yohannes, Carolyn Greig, Justine Davies.

**Data curation:** Theogene Uwizeyimana, Aimable Uwimana, Collins Fred Inkotanyi, Dina Goodman-Palmer, Samuel Ntaganira, Leslie Kanyana, Maria Lisa Odland, Sandra Agyapong-Badu, Lisa Hirschhorn, Tsion Yohannes, Carolyn Greig, Justine Davies.

**Formal analysis:** Theogene Uwizeyimana, Aimable Uwimana, Collins Fred Inkotanyi, Dina Goodman-Palmer, Samuel Ntaganira, Leslie Kanyana, Maria Lisa Odland, Sandra Agyapong-Badu, Lisa Hirschhorn, Tsion Yohannes, Carolyn Greig, Justine Davies.

**Funding acquisition:** Carolyn Greig, Justine Davies.

**Investigation:** Carolyn Greig, Justine Davies.

**Methodology:** Theogene Uwizeyimana, Aimable Uwimana, Dina Goodman-Palmer, Samuel Ntaganira, Maria Lisa Odland, Sandra Agyapong-Badu, Lisa Hirschhorn, Tsion Yohannes, Carolyn Greig, Justine Davies.

**Project administration:** Theogene Uwizeyimana, Aimable Uwimana, Dina Goodman-Palmer, Maria Lisa Odland, Sandra Agyapong-Badu, Lisa Hirschhorn, Tsion Yohannes, Carolyn Greig, Justine Davies.

**Resources:** Dina Goodman-Palmer, Carolyn Greig.

**Supervision:** Carolyn Greig, Justine Davies.

**Visualization:** Dina Goodman-Palmer, Maria Lisa Odland, Sandra Agyapong-Badu, Lisa Hirschhorn, Tsion Yohannes, Carolyn Greig, Justine Davies.

**Writing – original draft:** Theogene Uwizeyimana, Aimable Uwimana, Collins Fred Inkotanyi, Dina Goodman-Palmer, Samuel Ntaganira, Leslie Kanyana, Maria Lisa Odland, Sandra Agyapong-Badu, Lisa Hirschhorn, Tsion Yohannes, Carolyn Greig, Justine Davies.

**Writing – review & editing:** Theogene Uwizeyimana, Aimable Uwimana, Collins Fred Inkotanyi, Dina Goodman-Palmer, Samuel Ntaganira, Leslie Kanyana, Maria Lisa Odland, Sandra Agyapong-Badu, Lisa Hirschhorn, Tsion Yohannes, Carolyn Greig, Justine Davies.

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
