## [Decision Letter · Decision Letter 0]

11 Sep 2023

PONE-D-23-19881Priorities and barriers for ageing well; results from stakeholder workshops in rural and urban Rwanda.PLOS ONE

Dear Dr. Goodman-Palmer,

Thank you for submitting your manuscript to PLOS ONE. After careful consideration, we feel that it has merit but does not fully meet PLOS ONE’s publication criteria as it currently stands. Therefore, we invite you to submit a revised version of the manuscript that addresses the points raised during the review process.

We look forward to receiving your revised manuscript.

Kind regards,

Timothy Omara, PhD

Academic Editor

PLOS ONE

Reviewers' comments:

Reviewer's Responses to Questions

**Comments to the Author**

1. Is the manuscript technically sound, and do the data support the conclusions?

Reviewer #1: Partly

Reviewer #2: Yes

2. Has the statistical analysis been performed appropriately and rigorously? 

Reviewer #1: N/A

Reviewer #2: N/A

3. Have the authors made all data underlying the findings in their manuscript fully available?

Reviewer #1: No

Reviewer #2: Yes

4. Is the manuscript presented in an intelligible fashion and written in standard English?

Reviewer #1: Yes

Reviewer #2: Yes

5. Review Comments to the Author

Reviewer #1: I have tremendously enjoyed this report, and I think it is making a great contribution to our knowledge that intercultural and contextual differences in perception of public healthcare deserve much more careful consideration.

The manuscript is well written, and concise. I do have a few concerns though:

While the nature of this research is qualitative, authors have not reported their findings with rigorous qualitative methods. This report pertains to observations from a workshop, serving as a semi-structured interview/focus group--but authors state that this was not a focus group and hence not recorded.

Specifically, I would have liked to learn more about the selective sampling process, guided by stakeholders. What pool of potential participants was available, what percentage were invited to participate; and how were those selected? A Table to indicate characteristics of each participant's occupation, disability, etc. would help.

Also, the dynamics through which individuals reached a consensus on the single priority needs to be explained . It is natural to expect disagreements existing (and if not, this should be reported too.) and the reader would learn from attention to such. (it's understood that the authors were blind to which participant said what.) A few quotations or specific examples can help.

It is not clear how stakeholder and older participants' responses were summarized (since they were not recorded). Who did the summarization? What were the criteria? The same applies to the process of categorization.

Were the four categories pre-determined? Were they implied in the questions? Were they deduced from the summaries? If so, were the responses open-coded? Please refer to a standard qualitative methodology to describe your approach, so others can replicate the effort in other contexts.

Please rephrase statements such as "we tended towards a deterministic approach" to not confuse readers with philosophical connotations. I think in explaining this "deterministinc approach" it is important to orient the user about existing healthcare infrastructure for older care in Rwanda--what "externalities" exist?

Reviewer #2: Thank you for letting me review this promising manuscript, which I truly enjoyed reading. I want to start with commending the authors for their work with listening to and highlighting the voices of the end users of a policy. I also enjoyed reading about how this was done, which I think will be useful for other researchers in the field.

In order for this paper to be published, I have some comments that I suggest the authors first consider.

1. The objectives are to “identify and compare the needs and priorities of older people and other stakeholders involved in caring for them in rural and urban areas of Rwanda”. I appreciate the comparison as this identifies the unique perspective of older people and provides a great argument for involving them, and not just stakeholders. However, in the results (and very clearly in the Conclusions), more focus lies on the comparisons that the identification. To read about the actual ‘needs and priorities’, the reader is referred to figures and tables, which are useful to display all information in detail, but as I was expecting to learn more about needs and priorities, I find myself lacking an overview of these. I suggest rewriting the result to add more overview of the needs expressed in the workshops.

2. The stakeholder group (under ‘Study population’) needs to be described more in detail. The table in the appendix is useful, however as a reader not familiar with the context I can’t get a good picture of what kind of organisations these are. It needs to be more clearly described in the text.

3. The “socio-ecological model” used is not properly referenced or introduced when first mentioned. Please add this.

4. Line 129: “results were translated after prioritization” – what does this mean?

5. Line 175-176: “Authors did not have access to information…” This statement is difficult to understand. I assume you saw their faces and heard their voices in the workshop, which would be information to identify someone. Did you mean that you did not record and/or save any personal information?

6. A technicality around Figure 3: In several places, two sections of text are placed in the same space, making it difficult to read.

6. PLOS authors have the option to publish the peer review history of their article (what does this mean?). If published, this will include your full peer review and any attached files.

Reviewer #1: No

Reviewer #2: No

---

## [Author Response · Author response to Decision Letter 0]

6 Nov 2023

Editor’s questions to reviewers

3. Have the authors made all data underlying the findings in their manuscript fully available?

The PLOS Data policy requires authors to make all data underlying the findings described in their manuscript fully available without restriction, with rare exception (please refer to the Data Availability Statement in the manuscript PDF file). The data should be provided as part of the manuscript or its supporting information or deposited to a public repository. For example, in addition to summary statistics, the data points behind means, medians and variance measures should be available. If there are restrictions on publicly sharing data—e.g. participant privacy or use of data from a third party—those must be specified.

Reviewer #1: No

Reviewer #2: Yes

Response: Reviewer 2 is correct, all the data that were collected in the workshops form part of the paper, they are thus available to all readers without restriction.

Reviewer Comments to the Author

Reviewer #1: I have tremendously enjoyed this report, and I think it is making a great contribution to our knowledge that intercultural and contextual differences in perception of public healthcare deserve much more careful consideration.

The manuscript is well written, and concise. 

Response: Thank-you for your positive comments

I do have a few concerns though:

While the nature of this research is qualitative, authors have not reported their findings with rigorous qualitative methods. This report pertains to observations from a workshop, serving as a semi-structured interview/focus group--but authors state that this was not a focus group and hence not recorded.

Response: We apologise if the text in the article was misleading and led to the expectation that this was a formal qualitative study involving focus group discussions which were recorded, transcribed, and analysed in the study. It was not; the methodology was based on a nominal group technique, delivered in a workshop setting with plenary and roundtable discussions, to reach consensus on lists of priorities. The outputs were lists of agreed priorities. We have used the nominal group technique as it is recognized to be a valuable method for engaging discussions within a group and getting rapid agreement on the relative importance of issues and solutions. We have gone through the manuscript and removed instances where we had introduced terms like “qualitative” and “focus groups”, which may introduce confusion. We have also added references for the nominal group technique method, to facilitate understanding.

Specifically, I would have liked to learn more about the selective sampling process, guided by stakeholders. What pool of potential participants was available, what percentage were invited to participate; and how were those selected? A Table to indicate characteristics of each participant's occupation, disability, etc. would help.

Response: Each stakeholder was asked to nominate elderly individuals from the list of those engaged in their services or who were known in their communities. We requested that stakeholders nominate older participants to represent a broad range of characteristics (e.g., those living with disabilities and chronic diseases, farmers, and those who worked in public or private services before retirement). Elderly people nominated by stakeholders were then listed in random order by district and cell (a subdivision of a sector, which is a subdivision of a district), and were called by phone until the required number confirmed their availability. From each cell at least 2-3 people were selected such that a total of around 15 elderly individuals were invited in each study area (Kigali and Burera), These participants were retirees, but they represented a diverse set of characteristics, as detailed in lines 121 – 123 of the manuscript. 

We have included the above text in the manuscript to make this clearer.

Also, the dynamics through which individuals reached a consensus on the single priority needs to be explained. It is natural to expect disagreements existing (and if not, this should be reported too.) and the reader would learn from attention to such. (it's understood that the authors were blind to which participant said what.) A few quotations or specific examples can help.

Response: The nominal group technique focuses on the nomination of ideas by participants at roundtables. Discussion was encouraged in the roundtable and plenary sessions prior to prioritizing the lists, such that the prioritization was informed by the prior discussion. Hence, disagreements on the selected priorities or their orders are uncommon. Where disagreements occurred, facilitators worked with the discussion group to improve agreement. The final list was selected by voting by all participants, and disagreement on the order after voting is unusual in nominal group technique methods. 

We have added this text to the manuscript to improve clarity around resolution of disagreements. 

It is not clear how stakeholder and older participants' responses were summarized (since they were not recorded). Who did the summarization? What were the criteria? The same applies to the process of categorization.

Response: Please also see responses above. The methodology used was nominal group technique, which involves discussion to orient individuals to the issues under consideration followed by the capture of lists of priorities. The facilitators captured the lists from the participants on round tables, then after discussion, the participants were asked to rank their list. The ranking was done by facilitated discussion, then there was voting in plenary on the lists put forwards by round tables. There was some summarization which occurred in the combination of responses from the multiple roundtables such that the lists presented for voting were free of duplicates. This was done by agreement of the facilitators of the meeting. 

We have added clarity to the manuscript that prior to voting in plenary, the facilitators of the meeting discussed the lists provided by roundtables and remove duplicate ideas.

The text now reads: “Following the discussion, the facilitators combined the top 4 responses from each of the small groups after discussing amongst the facilitator group which were duplicate responses and removing these. All stakeholders then voted for the top 4 priorities for each question; facilitators recorded these priorities.”

Were the four categories pre-determined? Were they implied in the questions? Were they deduced from the summaries? If so, were the responses open-coded? Please refer to a standard qualitative methodology to describe your approach, so others can replicate the effort in other contexts.

Response: Yes, the four categories were predetermined and implied in the questions. As per above responses, this was not a qualitative study, and no coding was done.

Please rephrase statements such as "we tended towards a deterministic approach" to not confuse readers with philosophical connotations. I think in explaining this "deterministic approach" it is important to orient the user about existing healthcare infrastructure for older care in Rwanda--what "externalities" exist?

Response: We apologise, the manuscript should have read that we used the philosophy of determinism in our analysis, when categorizing responses in the socio ecological model. We have adjusted the text changing “we tended towards a deterministic approach” and to “we were guided by philosophy of determinism” It is challenging in practice to know the balance between whether behavior is medicated by individual’s agency or externalities. The philosophy of determinism suggests that most events or behavior are governed by externalities. This philosophy has gained traction in individual health, especially in high income countries and in particular around making healthy choices. Drivers of choices in Rwanda, whether predominately through agency or externalities has been understudied. However, on discussion, the authors agreed that in many cases, the existence of externalities was more likely to influence decisions and choices than agency, for the issues under discussion. 

We have included the above text in the limitations of the manuscript. 

Additionally in the discussion, we have added some context on healthcare access in Rwanda, as follows: Older people in Rwanda have the right to health and are supported in acquiring medical insurance like Mutuelle de Sante, receiving social support (VUP), and accessing the pension system through the Rwanda Social Support Board (RSSB). These elements were considered existing externalities in our classification using the socio-ecological model. 

Reviewer #2: Thank you for letting me review this promising manuscript, which I truly enjoyed reading. I want to start with commending the authors for their work with listening to and highlighting the voices of the end users of a policy. I also enjoyed reading about how this was done, which I think will be useful for other researchers in the field.

Response: Thank-you very much for these positive comments

In order for this paper to be published, I have some comments that I suggest the authors first consider.

1. The objectives are to “identify and compare the needs and priorities of older people and other stakeholders involved in caring for them in rural and urban areas of Rwanda”. I appreciate the comparison as this identifies the unique perspective of older people and provides a great argument for involving them, and not just stakeholders. However, in the results (and very clearly in the Conclusions), more focus lies on the comparisons that the identification. To read about the actual ‘needs and priorities’, the reader is referred to figures and tables, which are useful to display all information in detail, but as I was expecting to learn more about needs and priorities, I find myself lacking an overview of these. I suggest rewriting the result to add more overview of the needs expressed in the workshops.

Response: We have added some general description of the results from each group, in addition to the comparisons. 

We have added the following text: “The results from both workshops, highlighted that it is important to older people to prioritise physical and mental health, improve the health system and provide access to healthcare services, and provide safe housing. Stakeholders in both workshops felt it was important for older people to have a healthy and close-knit family, be financially stable, live in a peaceful country, preserve culture and tradition and own properties. The priorities that need to be addressed, put forwards by stakeholders, include creating advocacy for the elderly at a national level, establishing an elderly monitoring office, the provision of a healthcare insurance scheme and a pension plan, and being cared for by government and family. The main enablers that older people put forwards to help them maintain health and well-being were good governance and security, improved local infrastructure, good nutrition, and ability to pay for medical services. The main obstacles that older people put forwards were difficulty accessing transport services, lack of food markets specifically for older people, inability to meet their basic needs due to poverty, and chronic illness.” 

2. The stakeholder group (under ‘Study population’) needs to be described more in detail. The table in the appendix is useful, however as a reader not familiar with the context I can’t get a good picture of what kind of organizations these are. It needs to be more clearly described in the text.

Response: The organizations invited as stakeholders included charitable, civil society, government, and religious groups that work closely with the elderly in Rwanda by providing services or care. For each type, at least 1-2 representatives were invited. Out of the 28 invited, 23 attended from both Burera and Kigali.

We have added this text in the manuscript to improve the clarity.

3. The “socio-ecological model” used is not properly referenced or introduced when first mentioned. Please add this.

Response: We have adjusted the referencing for the SEM and added some text for clarity when we first mention is as follows: “The social-ecological model (SEM) is a framework advanced by the World Health Organization. It aids in understanding the diverse influences on an individual's health and well-being, including Individual, Relationships, Environmental, and Social or Community levels.” The reference is provided; see reference number 11. 

4. Line 129: “results were translated after prioritization” – what does this mean?

Response: We conducted all the workshops and ranked the priorities from the discussions in Kinyarwanda (local language). The Rwanda team translated the priorities into English, and Rwanda and English-speaking authors discussed the translation to make sure that the meaning was clear in English and accurate in Kinyarwanda.

5. Line 175-176: “Authors did not have access to information…” This statement is difficult to understand. I assume you saw their faces and heard their voices in the workshop, which would be information to identify someone. Did you mean that you did not record and/or save any personal information?

Response: We reviewed the entire document and did not find the statement, “Authors did not have access to information…”. In the workshop, only four authors facilitated the discussions; however, no recordings were made, and no personal data were saved.

---

## [Decision Letter · Decision Letter 1]

14 Nov 2023

PONE-D-23-19881R1Priorities and barriers for ageing well; results from stakeholder workshops in rural and urban Rwanda.PLOS ONE

Dear Dr. Goodman-Palmer,

Thank you for submitting your manuscript to PLOS ONE. After careful consideration, we feel that it has merit but does not fully meet PLOS ONE’s publication criteria as it currently stands. Therefore, we invite you to submit a revised version of the manuscript that addresses the points raised during the review process.

We look forward to receiving your revised manuscript.

Kind regards,

Timothy Omara, PhD

Academic Editor

PLOS ONE

Journal Requirements:

Reviewers' comments:

Reviewer's Responses to Questions

**Comments to the Author**

1. If the authors have adequately addressed your comments raised in a previous round of review and you feel that this manuscript is now acceptable for publication, you may indicate that here to bypass the “Comments to the Author” section, enter your conflict of interest statement in the “Confidential to Editor” section, and submit your "Accept" recommendation.

Reviewer #1: All comments have been addressed

Reviewer #2: (No Response)

2. Is the manuscript technically sound, and do the data support the conclusions?

Reviewer #1: Yes

Reviewer #2: Yes

3. Has the statistical analysis been performed appropriately and rigorously? 

Reviewer #1: N/A

Reviewer #2: N/A

4. Have the authors made all data underlying the findings in their manuscript fully available?

Reviewer #1: Yes

Reviewer #2: Yes

5. Is the manuscript presented in an intelligible fashion and written in standard English?

Reviewer #1: No

Reviewer #2: No

6. Review Comments to the Author

Reviewer #1: Thank you for addressing the comments.

I think a bit more work will make your paper more poignant.

Thank you for explain your SEM and the determinism philosophy, but they need to be a bit more cohesively explained and tied into your research method which is nominal group technique. Use philosophy to explain your theoretical mode that inform the research you want to do and necessitates the research method you have chosen.

If determinism is the philosophical ground for this study, then it needs to be stated as such in the introduction, and then it need to be explained how the socioecological model fits within this philosophical framework, and THEN explain why the nominal group technique was suitable to test the socio-ecological model that was informed by your determinism philosophy.

There are also typos such as behavior is mediCated--created by autocorrect.

Reviewer #2: Thank you for inviting me to re-review this manuscript. I want to acknowledge that the authors have indeed made improvement to the manuscript, and that this has made it clearer and more aligned with the stated aim.

I suggest a few minor adjustments to get this manuscript in order for publication.

• The aims in the abstract and in the end of the introduction are different, not just in phrasing but in a key concept: whether you aimed to compare or not. This needs to be aligned.

• There is a need to oversee the writing in the revised parts of the manuscript (perhaps using ‘track changes’ has been difficult?). For example, ‘Discussion’ is written two times and some sentences are written in a way that they repeat themselves. Another example is lines 132-134 which need to be fixed, the sentence is not understandable.

• I get what the ‘top responses’ mean as there was a prioritization process, but what is the ‘first four responses? There is a mix-up between the concepts in the manuscript (for example line 245 refer ‘the first four’ in table 1, but this table shows ‘top fours priorities’) which makes me assume that it is actually the same thing. These terms need to be overseen.

• This has not been amended since I first reviewed this manuscript: In Figure 3, in several places, two sections of text are placed in the same space, making it difficult to read.

Thank you again for a great manuscript!

7. PLOS authors have the option to publish the peer review history of their article (what does this mean?). If published, this will include your full peer review and any attached files.

Reviewer #1: No

Reviewer #2: No

---

## [Author Response · Author response to Decision Letter 1]

29 Dec 2023

Dear Editors and reviewers,

Thank-you very much for taking the time to review our paper. Thank you also for the positive comments on it. We have taken all comments into consideration when revising the paper, and our responses are documented below.

We look forward to hearing from you in due course.

Kind regards

Theogene on behalf of the authors

Reviewer Comments to the Author

Reviewer's Responses to Questions

Comments to the Author

1. If the authors have adequately addressed your comments raised in a previous round of review and you feel that this manuscript is now acceptable for publication, you may indicate that here to bypass the “Comments to the Author” section, enter your conflict of interest statement in the “Confidential to Editor” section, and submit your "Accept" recommendation.

Reviewer #1: All comments have been addressed.

Reviewer #2: (No Response)

2. Is the manuscript technically sound, and do the data support the conclusions?

Reviewer #1: Yes

Reviewer #2: Yes

3. Has the statistical analysis been performed appropriately and rigorously?

Reviewer #1: N/A

Reviewer #2: N/A

4. Have the authors made all data underlying the findings in their manuscript fully available?

Reviewer #1: Yes

Reviewer #2: Yes

5. Is the manuscript presented in an intelligible fashion and written in standard English?

Reviewer #1: No

Reviewer #2: No

6. Review Comments to the Author

Reviewer #1: Thank you for addressing the comments.

I think a bit more work will make your paper more poignant.

1. Thank you for explain your SEM and the determinism philosophy, but they need to be a bit more cohesively explained and tied into your research method which is nominal group technique. Use philosophy to explain your theoretical mode that inform the research you want to do and necessitates the research method you have chosen.

If determinism is the philosophical ground for this study, then it needs to be stated as such in the introduction, and then it need to be explained how the socioecological model fits within this philosophical framework, and THEN explain why the nominal group technique was suitable to test the socio-ecological model that was informed by your determinism philosophy.

Response to reviewer: 

Thank you, for the constructive comments. we have revised our manuscript to more explicitly state that our study is grounded in the philosophy of determinism. This is now clearly articulated in the introduction, highlighting how determinism views the aging process as significantly influenced by external societal and environmental factors and aligning with the socio-ecological model. The following statement was added in the manuscript: “To achieve this, we rely on the philosophy of determinism, which suggests that the conditions and experiences of persons – including older adults in Rwanda - are primarily shaped by external societal and environmental factors, rather than individual agency alone. To further categorize and analyze our findings, we use the socio-ecological model (SEM) as a framework. This model allows for a comprehensive understanding of the multiple influences on an individual's health and well-being, including individual, relational, environmental, and societal factors.” Line 99 – 109. 

In the methods section, we have also stated how nominal group technique is suitable to test SEM and determinism, by adding the following statement: “This method is consistent with our determinist approach, as it allows for the articulation of the influence of external societal and environmental factors on the individual, relational, environmental, and societal perspectives of both older people and stakeholders. Line 166 – 168. 

2. There are also typos such as behavior is mediCated--created by autocorrect.

Response to reviewer: 

We have gone through the manuscript and corrected any errors that we found, we have also made several small edits to improve the readability of the manuscript.

Reviewer #2: Thank you for inviting me to re-review this manuscript. I want to acknowledge that the authors have indeed made improvement to the manuscript, and that this has made it clearer and more aligned with the stated aim.

I suggest a few minor adjustments to get this manuscript in order for publication.

• The aims in the abstract and in the end of the introduction are different, not just in phrasing but in a key concept: whether you aimed to compare or not. This needs to be aligned.

Response to reviewer: This has been addressed. The aim in both places now reads: “This study aimed to identify and compare the needs and priorities of older people and other stakeholders involved in caring for them in rural and urban areas of Rwanda.”

• There is a need to oversee the writing in the revised parts of the manuscript (perhaps using ‘track changes’ has been difficult?). For example, ‘Discussion’ is written two times and some sentences are written in a way that they repeat themselves. Another example is lines 132-134 which need to be fixed, the sentence is not understandable.

Response to reviewer: This was addressed. Lines 156 -160 now read: “All communication was conducted in Kinyarwanda, the local language, and results were translated into English for analysis. Translations into English were done by the Rwanda team, and both Rwandan and English-speaking authors then discussed the translation to ensure that the meaning was clear in English and remained accurate in Kinyarwanda.” 

• I get what the ‘top responses’ mean as there was a prioritization process, but what is the ‘first four responses? There is a mix-up between the concepts in the manuscript (for example line 245 refer ‘the first four’ in table 1, but this table shows ‘top fours priorities’) which makes me assume that it is actually the same thing. These terms need to be overseen.

Response to reviewer: This has been addressed. The manuscript was revised to consistently use the terms “top four priorities” and “top four responses”, replacing “first four”. We hope that it is now clear.

• This has not been amended since I first reviewed this manuscript: In Figure 3, in several places, two sections of text are placed in the same space, making it difficult to read.

Response to reviewer: This has been addressed. Adjustments were made to the text size and positions to enhance readability.

Thank you again for a great manuscript!

Response to reviewer: Thank-you for your positive comment!

---

## [Editor Report · Decision Letter 2]

3 Jan 2024

Priorities and barriers for ageing well; results from stakeholder workshops in rural and urban Rwanda.

PONE-D-23-19881R2

Dear Dr. Goodman-Palmer,

We’re pleased to inform you that your manuscript has been judged scientifically suitable for publication and will be formally accepted for publication once it meets all outstanding technical requirements.

Kind regards,

Timothy Omara, PhD

Academic Editor

PLOS ONE
---

## [Editor Report · Acceptance letter]

21 Mar 2024

PONE-D-23-19881R2 

PLOS ONE

Dear Dr. Goodman-Palmer, 

I'm pleased to inform you that your manuscript has been deemed suitable for publication in PLOS ONE. Congratulations! Your manuscript is now being handed over to our production team.

Kind regards, 

on behalf of

Dr. Timothy Omara 

Academic Editor

PLOS ONE